# Efficient modification and preparation of circular DNA for expression in cell culture

Roman Teo Oliynyk [1,2 ✉] & George M. Church[1,3]

DNA plasmids are an essential tool for delivery and expression of RNAs and proteins in cell culture experiments. The preparation of plasmids typically involves a laborious process of bacterial cloning, validation, and purification. While the expression plasmids can be designed and ordered from the contract manufacturers, the cost may be prohibitive when a large number of plasmids is required. We have developed an efficient fully synthetic method and protocol that enables the production of circularized DNA containing expression elements ready for transfection in as little as 3 hours, thereby eliminating the bacterial cloning steps. The protocol describes how to take a linear double-stranded DNA fragment and efficiently circularize and purify this DNA fragment with minimal hands-on time. As proof of the principle, we applied Circular Vector expressing engineered prime editing guide RNA (epegRNA) in cell culture, and demonstrated matching and even exceeding performance of this method as compared to guides expressed by plasmids. The method's speed of preparation, low cost, and ease of use will make it a useful tool in applications requiring the expression of short RNAs and proteins.

[1] Department of Genetics, Harvard Medical School, Boston, MA, USA. [2] Department of Computer Science, University of Auckland, Auckland, New Zealand. [3] Wyss Institute for Biologically Inspired Engineering at Harvard University, Boston, MA, USA. ✉email: roli573@aucklanduni.ac.nz

M any techniques exist for the preparation and delivery of RNAs and proteins for transient expression in cell cultures[1]. In this study, we present a new and efficient preparation method. As a demonstration, we will describe the use of our method in the gene editing application context. However, since the principle behind this method is generic, this method can be useful for a variety of purposes and applications that require the expression of RNAs and proteins. Although we will discuss the specific implementation for gene editing and compare our method with some existing preparation techniques, the goal of this paper is to present a new approach to preparing the expression vectors in a laboratory environment—and not a more efficient gene editing method as such.

The three most customary approaches that facilitate gene editing when delivered into cells are: (I) single or multiple plasmids[2] encoding Cas9 and other supplemental proteins and implementations of guide RNA suitable for traditional CRISPR/ Cas9, base editing, prime editing or other methods[3–6]; (II) Cas9 mRNA and guide RNA[7]; (III) Cas9 protein complexed with guide RNAs[8]. A variety of delivery methods from the aforementioned gene editing constructs are currently being used. The choice of delivery method will depend on the specific requirements and the researcher's preferences, which can include electroporation[9,10], lipofection[11], direct physical transfection[12], vector delivery on polymer particles and microcarriers[13], and modalities of the aforementioned methods.

The approach (I), gene editing using plasmid transfection for transient expression is commonly used due to its conceptual simplicity, ease of handling, and the stability of plasmids. However, the preparation of plasmids is labor intensive and time consuming process that typically takes 2 days[2] (reference typical bacterial cloning steps in Supplementary Note 1).

We have developed an efficient fully synthetic method and protocol that enables the production of circularized DNA containing expression elements ready for transfection in as little as 3 h, thereby eliminating the bacterial cloning steps. Circularized DNA is resistant to exonuclease degradation in the cytoplasm[14]. Linear DNA lacks such stability, which is a significant barrier to utilizing it for gene delivery. Moreover, our method takes advantage of this difference in properties by using exonuclease to digest unreacted or misreacted linear DNA fragments, thereby purifying the circular DNA.

We determined that the practical range of the expression vector length is in between 450 and 950 bp, where it reaches up to 62% efficiency of converting input linear double-stranded DNA (dsDNA) into the circular expression vectors; the method could be also used with lower efficiency for somewhat longer fragments. For brevity, consistency, and to avoid confusion with the description of preparation steps and final product in this paper, we will refer to this circular DNA expression vector construct and method as a *Circular Vector*, capitalized and in italics.

To demonstrate that such circular DNA constructs can work as expression vectors, we applied this method to create circular DNA for expressing the engineered prime editing guide RNA (epegRNA)[5,15,16], and we successfully edited three genomic locations that were showcased by Chen et al.[15], resulting in a comparable editing efficiency.

## Results

The goal of this research was to develop and validate an efficient single tube preparation method for delivery and expression of RNAs and proteins in cell culture experiments. Circularization of the prime editing guides was performed following the protocol described in the Methods section, and schematically depicted in Fig. 1.

The relative yield of the reaction (see Fig. 2a), is defined in Eq. (1) by the ratio of the amount of circularized DNA after purification to the theoretical maximum possible yield when excluding the restriction enzyme sites and DNA end padding or PCR adapters (see Supplementary Fig. 1 and further description in the Methods section). The typical length of the circularized PEmax epegRNA expression construct would be between 450 and 500 bp, including the promoter and the sequence coding for pegDNA with the end RNA loop, as described in Chen et al.[15]. For this research, we chose a 452 bp *Circular Vector* coding for a single nucleotide substitution in HBB gene[15]. We designed a set of dsDNA with varying padding lengths composed of randomized dsDNA sequences to test how the yield depends on the input dsDNA length and circularization (ligation) step duration. Adding such DNA padding of 100, 300, 500, 880, and 1330 bp to a bare bones 452 bp *Circular Vector* resulted in 552, 752, 952, 1332, and 1782 bp *Circular Vectors* (see Supplementary Fig. 1). In addition, we tested a shortened 282 bp circularized structure that was too short to contain a functional expression vector in our implementation but was interesting from the perspective of verifying the behavior of potentially shorter constructs.

The relative yield of the protocol for the tested *Circular Vector* lengths and circularization durations is presented in Fig. 2a. Notably, the yield was highest for the circular DNA length range coinciding with the lengths of our prime editing vectors (at or above 450 bp). Excluding this high yield peak, the yield remained relatively constant and dropped off rapidly for lengths above 1000 bp.

The gel electrophoresis results show a picture containing exclusively circularized DNA in multiple bands (see Fig. 2b). As expected, in addition to single-unit circularization, some ligations resulted in duplicate, triplicate, and higher-order concatemers, which are schematically depicted in Fig. 2c. This hypothesis was confirmed by cutting out, purifying and sequencing individual bands, as well as re-running gel electrophoresis on DNA purified out of individual bands. In every instance Sanger sequencing confirmed the perfect match between the designs and the constructed *Circular Vectors*. The bulk of the circularization occurred within the first hour, exceeding 48% at a size of 452 bp and decreasing to 26% at 952 bp, thereby making it a practically useful method throughout this length range. Unsurprisingly, while yield increased even further with increased circularization/ligation duration, the proportion of higher-order combinations also increased with ligation duration (see gel images in Fig. 2b).

Analysis of the gel electrophoresis images using the National Institutes of Health's ImageJ software[17] showed that the higher band representation by DNA weight slightly increased with circularization duration (see Fig. 3a). This is particularly noticeable when comparing Band 1 (single circular DNA, blue line) versus the cumulative value of Bands 3 and higher (yellow line). Only at a *Circular Vector* size of 282 bp does simplex Band 1 represent less than Band 2 and less than the combined value of Bands 3 and higher of the vector units by weight in the reaction output.

The number of molar units is always relatively high for Band 1 since the molar representation is proportionate to band DNA weight *divided* by band number (see Fig. 3b). For a 1 h ligation duration in a *Circular Vector* between 450 and 950 bp in length, Band 1 makes a molar fraction of ~70%, with Band 2 representing another 20%, and the higher-order bands combined representing the remaining 10–15%. At 12 h of ligation, the Band 1 share shifts lower by ~6%, while Band 2 remains rather constant, and the combined higher bands' share increased by ~6%. This represents an increase in higher-order bands weight with minimal, if any, additional yield for the single-unit *Circular Vectors*.

Inversely, above 1000 bp, the higher bands combine to below 5% molar fraction. However, due to their low yield of 5–8%, we consider these longer *Circular Vectors* as not practically useful in

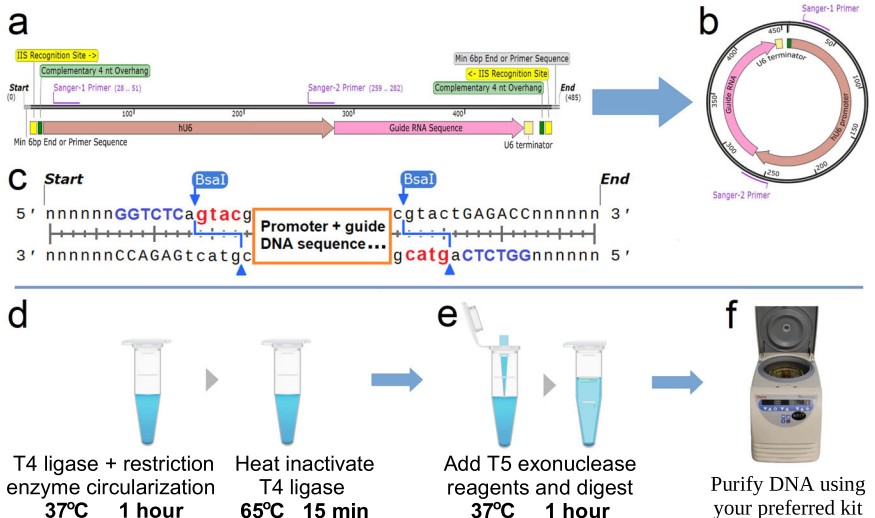

**Fig. 1 Circularization of dsDNA containing an expression element and the circularization protocol steps. a** Expression element containing type IIS restriction enzyme recognition sites with complementary overhangs on both ends of the DNA fragment. **b** Cuts by the restriction enzyme and ligation of the resulting overhangs form the circular DNA expression element. T5 exonuclease is used to digest all non-circularized DNA. **c** Schematic view of complementary 4nt overhangs (**gtac**) created by restriction enzyme cuts (BsaI **GGTCTC**), see Supplementary Fig. 1 for a detailed example of prime editing epegRNA coding. *Protocol steps:* (**d**) Pipette dsDNA template with T4 DNA ligase and type IIS restriction enzyme reagents, then program incubation at 37 °C for 1 h (or longer, if desired), followed by the heat activation of T4 DNA ligase at 65 °C for 15 min. If left unattended, set the program cycler to pause at 4 °C. **e** Add an equal volume of T5 exonuclease reagents to the same tube and digest at 37 °C for 1 h. **f** Purify the circularized DNA expression construct from the reaction mix using a DNA cleanup kit.

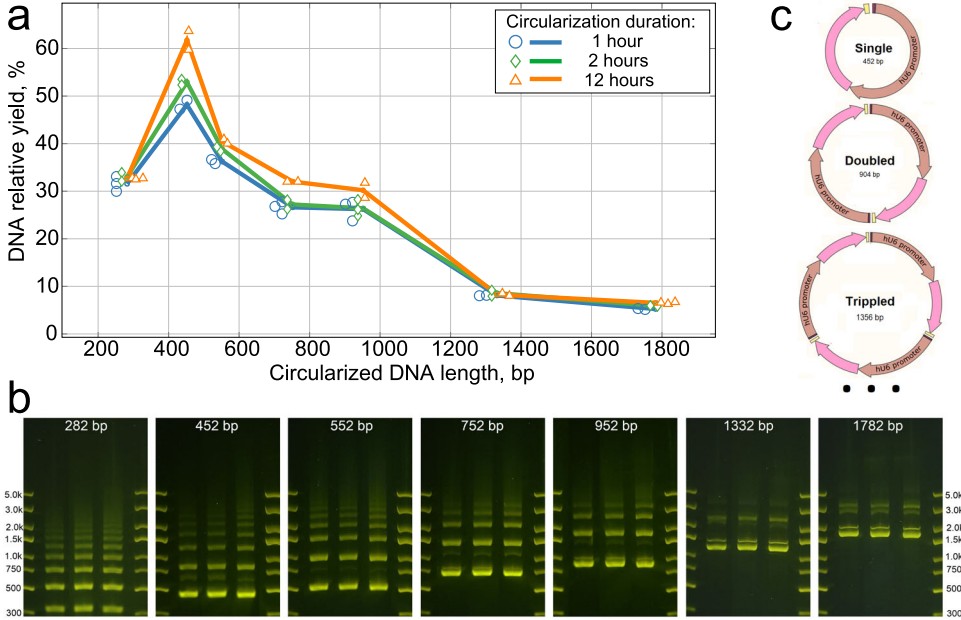

**Fig. 2 Circularization yield and concatemer proportions. a** Relative yield at a range of input DNA lengths at 1, 2 and 12 h of circularization. The line nodes represent the averages of $n = 2$–3 independent reaction samples; the dots show individual data samples for the nodes, horizontally offset for visibility. **b** Gel electrophoresis combination image. In each subplot, Lane 1 = 1 h, Lane 2 = 2 h, and Lane 3 = 12 h of circularization. The quantity of DNA was normalized to 300 ng per lane in subplots 282–952 bp. To avoid overload due to the small number of bands, DNA quantity was normalized to 150 ng per lane for the 1332 and 1782 bp subplots. The lanes in each subplot are a subset of lanes from a same single gel plate for each *Circular Vector* length, cropped only to reduce the number of lanes without any image enhancement; the complete gel photographs are available in the Supplementary Data. **c** Schematic diagram of self-circularization and the joining of repeated multiple fragments.

our method. We hypothesize that these changes in input DNA length for the yield and fraction distribution of joined *Circular Vector* concatemers—including the apparent spike in yield around 450 bp—are likely caused by a combination of DNA length and folding patterns affected by the reaction mix pH, ion influences, and the reaction temperature[18].

While the majority of circular vectors by number are comprised of single and double units, the higher-order concatemers are represented disproportionately by weight. For transfection, it is important to be able to calculate a molar proportion of the Cas9-carrying plasmid and guide plasmid. Plasmid transfection efficiency is always <100% of cells, and varies depending on cell

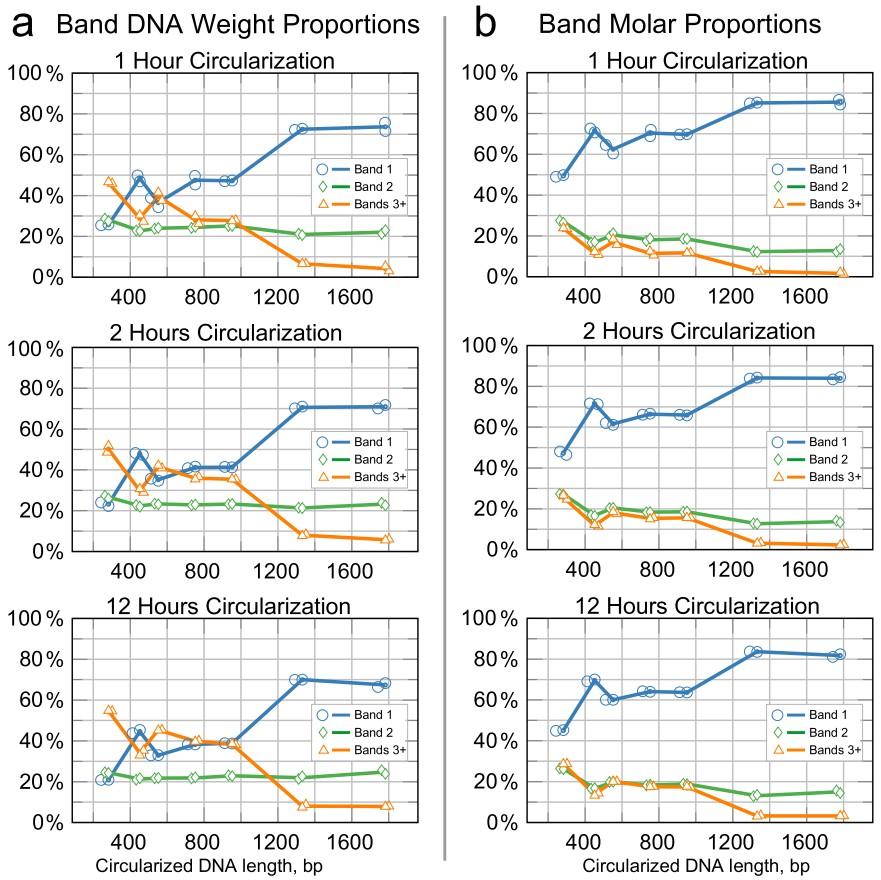

**Fig. 3 Characterization of the circularization product by ImageJ gel electrophoresis band brightness analysis.** We show the first, second, and the sum of third and higher bands corresponding to single, doubled, and the sum of the higher-order concatemers. The line nodes represent the averages of $n = 2$ independent reaction samples calculated from the gel electrophoresis band brightness; the dots show individual data samples for the nodes, horizontally offset for visibility. **a** Relative DNA weight is equal to relative band brightness. **b** Relative band molar count is counted as relative band brightness divided by band number (see also Eq. (2)).

type, size of plasmid(s) and transfection methods. Determining of the perfect ratio when co-transfecting two plasmids is laborious[19], and when two co-transfected plasmids differ in size, a molar excess of the smaller plasmid is customarily used, see for example Bosch et al.[20]. In our prime editing validation experiments, the *Circular Vector*s are five times smaller than the plasmids expressing epegRNA[16] that they substitute, while these epegRNA plasmids[16] are themselves about five times smaller than the matching Cas9 expressing plasmids[15] (see further discussion). Thus, we calculate a "molar multiplier" that indicates how much more *Circular Vector* by weight must be used when compared to the case of a pure single-unit *Circular Vector* using Equation (2). The molar multiplier values for 1 h and 12 h of ligation in Fig. 4a imply that a longer ligation duration results in a molar ratio increase due to a predominant increase in the higher band fraction.

Validation of the *Circular Vector* functionality was performed by prime editing of the single base substitutions in HEK293T cell culture at three genomic locations (i.e., HBB, PRPN, and CDKL5) that were used by Chen et al.[15] as a demonstration of improved efficiency of their PEmax method. We coded the guide design from a study by Nelson et al.[16], in conjunction with the Addgene plasmid #174828[15,21], which was modified with the addition of blasticidin resistance gene to facilitate an antibiotic selection. We used gel electrophoresis to purify the first band from the prepared HBB, PRPN, and CDKL5 *Circular Vectors*, thus obtaining the single expression module *Circular Vector*. We also prepared a set of same three expression units with added 500 bp of inactive

DNA padding before the U6 promoter, as described above and shown in Supplementary Fig. 1. The specifics of gene editing are noted in the Methods section Transfection, Handling, and Sequencing of Edited HEK293T Cells. We verified that the use of 8X molar ratio of the *Circular Vector* to Cas9 plasmids produced negligible difference in editing efficiency when compared to the 4X ratio, and thus we performed all the reported here genome editing experiments with the customary 4X molar excess.

The editing efficiency analysis performed with EditR[22,23] is summarized in Table 1 and illustrated in Fig. 4b. It is evident that the *Circular Vectors* are functional for epegRNA expression when used as a component of prime editing, using 4X molar excess of *Circular Vectors* to PE4max Cas9 plasmid. The editing efficiency was close to PE4max results reported by Chen et al.[15] for two out of three edited locations, and significantly exceeded the editing efficiency of Chen et al.[15] for the third location—the PRNP locus, thus validating the functional use of *Circular Vectors*. Notably, using the *Circular Vectors* as prepared by our protocol resulted in better editing efficiency than the purified single band *Circular Vector*, thus validating the advantage of using the mix containing a fraction of concatemers, rather than the single circularized units. The *Circular Vectors* with additional 500 bp padding displayed further 1–4% greater editing efficiency (marked in bold in Table 1).

We hypothesized two mechanisms leading to the results observed in Table 1: (A) Perhaps, very short–452 bp–circular units are interacting slightly less efficiently with the Polymerase III than longer circular structures[24], and doubling the length may

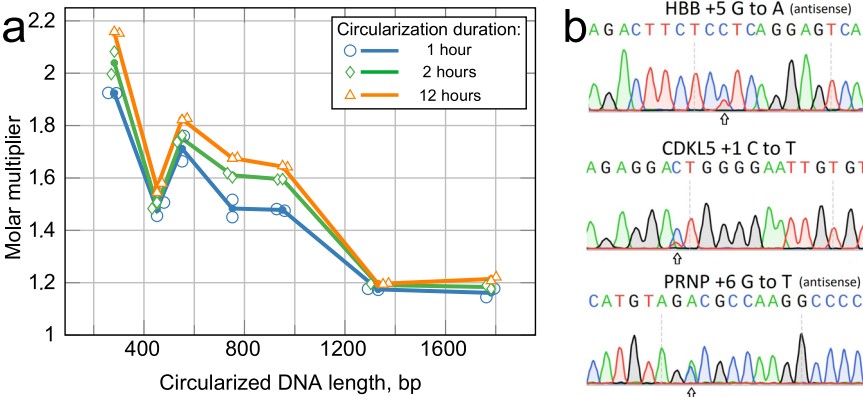

**Fig. 4 Effects of ligation duration and *Circular Vector* length on molar multiplier and Sanger sequencing validation of editing efficiency. a** Molar multiplier by length and circularization duration. Between 452 bp and 952 bp, a simple rule of thumb based on the distribution is a 1.5× multiplier after 1 h of circularization, and 1.7× for 2 h of circularization or longer. For example, if the Cas9 plasmid is 13,000 bp long and single circular vector is 452 bp long, then 1.0 $\mu$g of the main plasmid at molar ratio 1:1 will require $1.0 \times 452/13{,}000 = 0.035\,\mu$g $= 35$ ng. Multiplying by 1.5 will result in 52 ng of a vector rather than 35 ng for a 1:1 molar ratio based on a 1 h circularization reaction. The line nodes represent the averages of $n = 2$ independent reaction samples calculated from the gel electrophoresis band brightness; the dots show individual data samples for the nodes, horizontally offset for visibility. **b** Validation of the *Circular Vector* by prime editing HBB, CDKL5, and PRPN.

**Table 1 Comparison of prime editing efficiency with that of Chen et al.[15] for matching HBB, CDKL5, and PRPN locations.**

| Genomic location | Reference Chen et al.[a] (%) | | This research (%) | | |
| --- | --- | --- | --- | --- | --- |
| | PE2max | PE4max | Band 1 CV | CV | CV + 500 bp |
| HBB +5 G to A | 22.0 (0.5) | 35.2 (1.4) | 19.6 (2.1) | 23.8 (1.3) | **28.1** (0.4) |
| CDKL5 +1 C to T | 11.2 (0.6) | 25.2 (0.5) | 16.3 (0.4) | 20.7 (0.3) | **22.6** (0.5) |
| PRNP +6 G to T | 27.3 (0.8) | 35.7 (1.0) | 45.7 (1.2) | 54.6 (1.6) | **56.0** (0.6) |

[a]Prime editing efficiency estimated from Fig. 7D in Chen et al.[15]; values in () represent standard deviation of three data points. Column entries for our results stating *Band 1 CV* are editing with band 1 extracted using gel electrophoresis. *CV* denotes 452 bp *Circular Vectors* prepared by our protocol. *CV+500 pad* denotes added 500 bp padding to lengthen guide expression vectors to 952 bp. All entries in our results are the averages and standard deviations (in round brackets) from four independent samples provided by EditR analysis of Sanger sequencing traces, with exception of HBB Band 1 CV representing average of three samples. Edits performed on 24-well plates, using 4X guide molar excess (see Transfection, Handling, and Sequencing of Edited HEK293T Cells in Methods). Highest editing efficiency achieved is shown in bold.

make the interaction slightly more efficient, which may explain the observed small increase in efficiency between columns *CV* and *CV+500 pad*, which contained the additional 500 bp of inactive DNA padding. When maximum editing efficiency is required, the additional DNA padding can be easily incorporated in the *Circular Vector* design, as in Supplementary Fig. 1. (B) The larger increase in the editing efficiency between columns *Band 1 CV* and *CV* may be explained by ~1.5 times larger number of expression units per molar unit of the concatemer mix of *Circular Vectors* relative to the single circularized expression units; additionally the concatenated *Circular Vectors* are automatically longer. This also implies that extended circularization duration, resulting in a higher representation of concatenated *Circular Vectors*, may also slightly improve the editing efficiency at a fixed molar excess.

In summary, the above results demonstrated that 450–950 bp *Circular Vectors* prepared by using our protocol are suitable for delivery and expression of RNAs for the use as epegRNA expression vectors in prime editing in cell cultures.

## Discussion

The circularization method and protocol presented here describes how to prepare *Circular Vectors* that can be used as an efficient replacement for the guide plasmids in CRISPR, base, and prime gene editing in cell cultures. The main advantage of this method is that after designing, ordering, and receiving dsDNA from a commercial supplier, the product ready for transcription can be prepared for use within 3 h, with little hands-on time. Moreover,

the majority of hands-on time consists of the basic spin column kit DNA cleanup. If the amount of DNA received is low and PCR amplification is required, the process could take an extra hour to amplify the necessary quantity of input DNA.

There are two main limitations or concerns that we wish to discuss:

1. The amount of *Circular Vector* produced by a synthetic reaction. Typically, a single 50 $\mu$l reaction will produce 2.0–2.5 $\mu$g of product. Although the reaction can be scaled up with more reagents, in situations when a large amount of vectors is needed, the ability of bacterial cloning to produce milligrams of product in one maxi-prep will make it a better choice. In most situations, only a few edits are made with each guide, thus simplicity and speed of preparation favor our circularization method.

2. We would like to discuss the error rate inherent in the manufacturing process of DNA fragments. High fidelity is a strong suite of the bacterial cloning method. The *E. coli* DNA replication error rate as it clones a plasmid is estimated at $5 \cdot 10^{-10}$ per base pair[25]. Once an *E. coli* colony that contains a perfect plasmid is identified by sequencing and cloned, the chances of any plasmid containing an error are vanishingly small, provided that precautions are taken[26].

On the other hand, the error rate advertised by commercial DNA fragment manufacturers is quite high. Error rates of DNA fragments under 800 bp in length produced by three manufacturers are: Twist Bioscience 1/6253 bp, Thermo

Fisher Scientific 1/6757 bp, and Integrated DNA Technologies 1/6757 bp. Notably, both Twist Bioscience[27] and Thermo Fisher Scientific[28] claim that the true error rate for Integrated DNA Technologies gBlocks is much higher (1/2705 and 1/1329 respectively).

Let us consider the example of the Twist Bioscience manufacturing process—with an error rate of 1/6253 bp—and our 452 bp *Circular Vector* with 20 bp long targeting spacer. Thus, the chance of an error occurring in the spacer is approximately $20 \div 6,253 \approx 1/312$ for *Circular Vectors*. More likely than not, they will not find any target in genome, and could lead to off-target edit attempts in a very small proportion from this already small 1/312 fraction of potentially mismatched guides. This conclusion is supported by Chavez et al.[29] who demonstrated that edits can be tolerant to multiple mismatches between the gRNA and the inappropriately bound locus. Assuming that an error occurs within the 264 bp promoter area (see Supplementary Fig. 1), it will either leave it functional or render it non-functional. If the promoter is non-functional, it would have no effect on the accuracy of the edit. Similarly, if the error falls in the RNA scaffolding area, linker, extension, or end loop will most likely render the epegRNA non-functional, also with no effect on the accuracy of the edit. Moreover, *Circular Vectors* are typically supplied in excess of the Cas9 plasmid, and non-functional *Circular Vectors* will be backed up by functional ones.

Thus, the additional error rate introduced by *Circular Vectors* is expected to be orders of magnitude lower than that produced by the prime editing itself and to an even larger extent, CRISPR/Cas9 and base editing[3–5,15,30], because all of these methods inherently exhibit 4–15% level of off-target and mis-edits. The Sanger trace analysis conducted with EditR for our three edited genomic locations did not indicate noticeable off-target edits, which is consistent with the aforementioned reasoning that off-target edits with use of the *Circular Vectors* method, if any, will be orders of magnitude less frequent than successful edits.

In some cases, when multiple RNAs need to be expressed, a solution may be implemented with polycistronic guides[31]. U6 promoters express RNA without additional nuclear localization sequences present on a plasmid[16,32,33], and our validation experiments confirmed this for *Circular Vectors*. When implementing other promoters, the design—similarly to plasmids—may need to ensure nuclear localization[34,35].

There are certain situations where two or more RNAs must be expressed that require separate promoters, or multiple repeats are present in the desired dsDNA design. For example, the prime editing methods PE3 or PE5max[15] require a second RNA guide to produce an extra nick. Even more complex scenarios exist[6,36]. In such cases, multiple elements may be combined within one *Circular Vector*. Most manufacturers cannot make DNA fragments with long repeats (e.g., repeating promoters or pegRNA scaffolds). However, dsDNA can be ordered as separate fragments and performing the easy additional step of linear ligation can prepare a combined linear dsDNA that can be subsequently circularized by the *Circular Vector* method (see Supplementary Fig. 2 and Supplementary Note 3: *Linear Joining of dsDNA Fragments*). For example, prime editing PE3 and PE5[15] require two U6 promoters that allow the transcription of two RNA guides. The corresponding constructs are customarily inserted into single plasmid vector using Golden Gate or Gibson assembly, followed by bacterial cloning. In an even more complex scenario, many stages of assembly and bacterial cloning involve ~1 week of work and handling[36]. In principle, the joining step requires designing and ordering fragments that can be cut with an intermediate type IIS restriction enzyme (i.e., BbsI in the example presented in Supplementary Fig. 2). The joining can be accomplished in a couple of hours, with 1 h Step 1 reaction described in Methods, followed by DNA cleanup to discard short DNA end cut-offs. In our experience such linear ligation allows to join linear dsDNA with nearly 100% efficiency, with losses primarily being due to the cleanup using a spin column kit. This preliminary step can then be followed by circularization using the *Circular Vector* protocol, resulting in the entire process being completed many times faster than the many stages of bacterial cloning. In addition, in circumstances where a pure single unit *Circular Vector* is required, it can be extracted via gel electrophoresis.

Another important advantage of this method is its low cost. Our cost calculation example in Reagents, Materials and Costs section shows that the fixed cost of the linear dsDNA fragment ordered from a commercial supplier is $32.90 for a 452 bp *Circular Vector*. The reaction costs only $9.30. If PCR amplification is required, this adds $4.76 to the total and eliminates the need to reorder the input dsDNA fragments from a supplier. The protocol produces 2.0–2.5 µg of *Circular Vector*, which is sufficient for more than 10 edits on a 24-well plate (based on the quantities we used in Step-by-Step Protocol section) and ~40 edits on a 96-well plate. This compares favorably to bacterial cloning, where the complete cost from design to a transfection-ready plasmid is typically severalfold higher (in addition to much longer hands-on and overall time). It is even more expensive to order ready-made plasmids, as presented in Supplementary Note 5 on the example of pricing from two contract manufacturers, and yet more expensive to order prefabricated epegRNA.

## Conclusion

In this research, we presented a circularization method and protocol for the preparation of *Circular Vectors* that can be used in a variety of applications requiring the expression of short RNAs and proteins. For a 450 bp *Circular Vector* length, the efficiency of converting the input DNA into *Circular Vectors* was 48% at 1 h of ligation and up to 62% at 12 h of ligation. For a 950 bp length, this efficiency was 26% at 1 h of ligation and 30% at 12 h of ligation, with intermediate values for *Circular Vectors* between 450 and 950 bp in length. Longer reaction times corresponded with some improvement in *Circular Vector* yield. However, a short 1 h ligation step has the advantage of rapid and efficient 3 h start to end processing.

As proof of the principle, we applied Circular Vector expressing epegRNA in cell culture, and demonstrated matching and exceeding performance of this method as compared to plasmid delivery reported by Chen et al.[15]. Its speed, low cost, and ease of use will make this method another valuable tool in the gene-editing toolkit, while the reaction simplicity makes the protocol suitable for automated liquid-handling systems[37].

## Methods

**Conceptual introduction**. With this method, which consists of a single tube two-step reaction followed by a typical spin column-based DNA cleanup, a researcher can produce a batch of *Circular Vectors* in 3 h. The *Circular Vector* method simplifies making small variable targeting components that work in conjunction with an invariable large plasmid coding for Cas9 and/or other proteins and genes. This large plasmid can be cloned in large quantity in bacterial culture, extracted with one of the commercially available maxi-preps, and used for subsequent months or even years of experiments. Notably, the variable component for each editing target —the *Circular Vector*—is now a breeze to make.

Circularized DNA is resistant to exonuclease degradation in the cytoplasm[14]. Linear DNA lacks such stability, which is a significant barrier to utilizing it for gene delivery. Moreover, our method takes advantage of this difference in properties by using T5 exonuclease to digest unreacted or misreacted linear DNA fragments, thereby purifying the *Circular Vector* DNA.

**Table 2 Circularization reaction ingredients for 6.0 μg of input dsDNA in 50 μl and cycler schedule.**

**Reagents**

| Component | Volume (μl) |
|---|---|
| dsDNA, concentration ≥180 ng/μl | per concentration to 6.0 μg |
| T4 DNA Ligase Buffer (10X B0202S) | 5.0 |
| T4 DNA Ligase (NEB M0202L) | 3.0 |
| Bsai-HF2 (NEB R3733L) | 3.0 |
| ATP (10mM NEB P0756S) | 5.0 |
| Nuclease-free $H_2O$ (Invitrogen 10977015) | as needed to 50.0 μl |

**Cycler schedule**

| Step | Temperature | Time |
|---|---|---|
| Circularization | 37 °C | 60 min |
| T4 DNA ligase inactivation | 65 °C | 15 min |
| Pause | 4 °C | indefinitely |

Our circularization method for the CRISPR guide expression vectors requires an initial DNA sequence design, which can be created using the researcher's preferred tools (we used SnapGene 6.0.7), followed by ordering dsDNA fragments from a commercial manufacturer.

The prefabricated dsDNA should—at a minimum—contain a promoter (U6, in our example), followed by a DNA segment coding for epegRNA with a terminator[38] (see Fig. 1a and the detailed map in Supplementary Fig. 1). The epegRNA can be designed using a published tool suited to this purpose[15,16,39]. Two matching type IIS restriction enzyme recognition sites create complementary 4 nt overhangs near both ends of the dsDNA fragment. The *Circular Vector* can be conveniently validated with Sanger sequencing (see Fig. 1b). Placing two primers, Sanger-1 and Sanger-2, near the opposite locations of the circular structure has proven to be efficient and reliable, covering most of the circular structure with the overlap that includes the opposing primer near the middle of the FASTA sequence. If the whole plasmid sequencing is preferred, companies performing such sequencing—for example, Plasmidsaurus (www.plasmidsaurus.com)—require a minimum of 2000 bp circle, however techniques based on rolling circle amplification, like one developed by Octant[40] may be able to sequence short circular dsDNA.

The matching overhangs created by the restriction enzyme cuts are designed to provide a high-fidelity ligation[41], which is also aided by the absence of any other overhangs that would be present in a typical Golden Gate assembly (see Fig. 1c). The 'nnnnnn' sequences on Fig. 1c represent the minimum of six base pairs required on the back end of the restriction enzyme. Some manufacturers (e.g., Twist Bioscience) prefer to deliver their DNA fragments with end adapters that serve as well-designed PCR amplification primers and simultaneously eliminate the need to incorporate end sequences.

**Step-by-step protocol**. We aimed to make the method of *Circular Vector* production user-friendly and rapid. Thus, we primarily used reaction enzymes and buffers based on commercial kits, with only a few additional reagents. If desired, the kits may be substituted with individual reagents. If a sufficient amount of dsDNA is received from a commercial manufacturer, it may be used directly in the reaction; otherwise, it may need to be amplified by PCR to obtain the desired quantity. The protocol for the preparation of the circularized DNA vectors consists of three steps:

**Step 1: Circularization of the source DNA** (Fig. 1d). Mix the reaction reagents listed in Table 2 by pipetting, preferably on ice, into a 200 μl PCR tube or another suitable tube with at least double the Table 2 reaction volume. The extra volume will be required for the addition of the digestion reagents in Step 2. Place the tube in a thermal cycler and run the reaction as described in Table 2 cycler schedule.

**Step 2: Digesting all remaining linear DNA** (Fig. 1e). Prepare the T5 exonuclease reaction mix as listed in Table 3 and mix it in 1:1 proportion into the tube containing the reaction mix from the previous step. For example, if a 50 μl reaction was performed in Step 1, 50 μl will need to be added. Program and run the thermal cycler as listed in Table 3 cycler schedule.

**Step 3: Cleanup of the circularized DNA** (Fig. 1f). Use a spin column kit such as *QIAquick PCR Purification Kit*, or your preferred method.

The above protocol steps are sufficient for a practical application of the method. For in depth explanation of the steps and reasoning behind them read the following section.

**Protocol Implementation Details**.

1. **The restriction enzyme digestion and ligation** reaction is followed by 15 min of heat inactivation to end the ligation reaction (see Fig. 1D). The constant 37 °C reaction temperature was chosen to minimize mismatch ligation[42,43]. T4 DNA ligase inactivates at 65 °C. Since the restriction enzyme was chosen with a higher inactivation temperature, the restriction enzyme will remain active and slightly assist in the following digestion step. The concentrations of ligase and restriction enzyme were experimentally determined to work better with a slightly higher concentration than is typical in Golden Gate assembly. This higher concentration serves two purposes: it facilitates rapid processing, and, more importantly, the restriction enzyme rapidly cuts to free overhangs on both ends of a DNA fragment, thereby allowing the self-ligation of these overhangs on the same DNA fragment and limiting the proportion of multiple DNA fragments ligating on each other (see further discussion in the Results section). The concentrations of input linear dsDNA were tested in a range from 7.5 to 120 ng/μl and produced indistinguishable circularized DNA yield and composition. A concentration of 120 ng/μl results in the input of 6.0 μg DNA per 50 μl reaction, which was considered optimal from handling and reagent cost perspectives. The addition of ATP may be optional for a 1 h ligation duration, with increasing ATP concentrations from 1 mM provided by the NEB T4 DNA ligase buffer to 2 mM showing a slight improvement in yield, particularly at longer ligation durations. This may be due to the exhaustion of ATP with a high concentration of ligated DNA, as well as ATP inactivation at longer reaction durations. The 2 mM ATP concentration is well within the optimum range for T4 DNA ligase mixes[44], and the ATP reagent is inexpensive. The restriction enzyme substitution to a different type IIS enzyme may be required in cases where the sequence coding for epegRNA includes the BsaI recognition site (for an example of using BbsI in a slightly different scenario, see Supplementary Fig. 2).

2. **Digestion of non-circularized DNA** (Fig. 1e). This step consists of adding an equal amount of the T5 DNA exonuclease reagents to the ligation reaction. For example, 50 μl of exonuclease reaction mix is added to 50 μl of ligation reaction output, resulting in a comfortable level of 100 μl of reaction mix in a typical 200 μl Eppendorf PCR tube thermocycler (larger tubes and dry baths, etc. may be used if desired). The concentration of T5 DNA exonuclease was validated as sufficient to completely digest all linear DNA in under 1 h for all input DNA lengths tested in this research. We found that adding only T5 exonuclease to the ligation reaction resulted in a very slow digestion rate due to the absence of potassium ions. The role of NEBuffer 4 is to provide potassium anions that accelerate exonuclease reactions. While the concentration of potassium in the resulting mix is half that of the exclusively NEBuffer 4 reaction mix, it remains close to the optimum concentration for cleaving reactions[45]. The optimal balance between exonuclease and endonuclease activity for T5 exonuclease is at the reaction pH 7.9–8.0[46]. This is the pH of NEBuffer 4, while the ligation buffer pH is 7.5. Thus, a suitable amount of Tris base was added to increase pH to this range, as presented in Table 2 (main article). Notably, it is not recommended to excessively increase pH for the T5 exonuclease reaction mix. When we tested pH increases above 8.5, all DNA—including the circularized DNA—was completely and rapidly digested.

3. **Cleanup of the circularized DNA**. This step involves using a quality DNA cleanup kit to extract *Circular Vector* DNA of sufficient concentration and purity. We used a spin column kit, which was selected as described below. With a large number of such products available, the choice of kit is up to the researcher's preference; however, the yield may vary. We aimed to select a simple spin column-based extraction kit that would provide a consistently high yield, a low level of impurities, and a sufficiently high DNA concentration for accurate and consistent measurements. This purity was important for the direct use of the *Circular Vectors* in cell culture transfection and for the accuracy of concentration and yield measurements reported by this study. A number of kits were reviewed, and three kits were then tested and evaluated for their efficiency in DNA extraction and low chaotropic salt contamination: the Takara Bio *NucleoSpin Gel and PCR Cleanup* 740609.50[47], QIAGEN *QIAquick PCR Purification Kit* 28106[48], and NEB *Monarch PCR & DNA Cleanup Kit* T1030S[49].

**Table 3 T5 exonuclease reaction ingredients in 50 μl and cycler schedule.**

**Reagents**

| Component | Volume (μl) |
|---|---|
| NEBuffer 4 (10X B7004) | 5.0 |
| T5 Exonuclease (NEB M0663L) | 4.0 |
| Tris Base 200mM (BB-2686)[a] | 2.0 |
| Nuclease-free $H_2O$ (Invitrogen 10977015) | as needed to 50.0 μl |

**Cycler schedule**

| Step | Temperature | Time |
|---|---|---|
| Digesting linear DNA | 37 ℃ | 60 min |
| Pause | 4 ℃ | indefinitely |

[a]Tris Base 200 mM is a 1:4 dilution of Tris Base (1 M, pH > 10.0) in water.

Based on a validation test with the same set of PCR products, the QIAquick kit was chosen based on it having the best recovery of input to output DNA when compared to the other two kits (20% better than NucleoSpin and 30% better than NEB) and exhibiting a low amount of captured chaotropic salts when testing a blank PCR product cleanup without any input DNA. In addition, even though the QIAquick kit rates 30 μl as a minimal elution volume, it performs with a good compromise of yield and higher DNA concentrations and shows excellent purity when eluting with volumes below 20 μl, which was preferable for our prime editing validation experiments. The QIAquick PCR Purification Kit has proven very reliable and consistent, with only one outlier that had approximately half the yield observed in the whole set of experiments. This outlier result may have been due to a defect in the spin column; thus, the sample was discarded.

The QIAquick kit includes an optional pH indicator that is recommended to add to the binding buffer due to its importance in maximizing DNA yield. The indicator reagent allows us to verify whether the mix pH is optimal for binding. Notably, we determined that pH adjustment was required via the addition of one or more 10 μl of 3M pH = 5.2 sodium acetate, as recommended by QIAGEN[48].

Special attention was given to minimizing pipetting losses. The amount of input DNA was adjusted to result in output product DNA in the range of 2–4 μg, which provided a sufficient amount of the final DNA product for measurement and testing. In extreme cases, the amount of input dsDNA for the 1782 bp *Circular Vector* had to be scaled up to 27 μg, which still resulted in only a 1.3–1.7 μg yield, while an input DNA amount of 6–9 μg was sufficient in 450–950 bp range. Such low yield, which would require high DNA input and the use of large quantities of reagents to produce sufficient product, indicates that long *Circular Vectors* are impractical to synthesize. We preferably aimed to achieve a concentration of output DNA well over 80 ng/μl, which would result in high-purity DNA based on 260/230 and 260/280 ratios. This was additionally aided by performing an extra wash step when using the purification kits, which resulted in consistently pure DNA output with a minimal potential sacrifice in yield. The DNA concentration measurements were primarily performed on a NanoDrop Eight Spectrophotometer (ThermoFisher Scientific) and repeated on a DS-11 Series Spectrophotometer/Fluorometer (DeNovix). Notably, the measured concentrations were closely matched between these two devices.

Extraction of Band 1 for testing gene editing performance of the pure single *Circular Vector* with no duplicates was performed by DNA separation by electrophoresis on 1% E-Gel EX Agarose Gels (Invitrogen). The *Circular Vector* DNA was cleaned using Monarch DNA Gel Extraction Kit (T1020S) with one extra wash step.

**Equations used in data analysis.**

1. **The maximum possible yield ratio of the circularization reaction** is calculated by dividing the circularized DNA length, as determined by the restriction enzyme cut sites based on input dsDNA length. Typically, a type IIS restriction enzyme requires six base pairs or more between it and the end of the dsDNA strand to ensure efficient cutting. The restriction enzyme recognition site itself, an offset between the recognition site and the cut location, and one length of the overhang are also discarded. For our handling, it was convenient to retain the Twist Bioscience end adapters, which are 22 bp in length and can be used with Twist Bioscience primers for the PCR amplification of these DNA fragments. In this case, the maximum possible yield ratio denominator is the length of the dsDNA ordered from Twist Bioscience plus 44 bp. All of the elements can be formally accounted for by the following equation:

$$R = \frac{L_c}{L_c + 2 * L_{enz\_site} + 2 * L_{enz\_offset} + L_{cut\_length} + 2 * L_{end\_pad}}, \quad (1)$$

where $R$ is the maximum yield ratio, $L_c$ is the length of the circularized DNA, $L_{enz\_site}$ is the recognition enzyme site, $L_{enz\_offset}$ is the offset between the enzyme recognition site and the beginning of the cut, $L_{cut\_length}$ is the length of the cut (typically 4 bp), and $L_{end\_pad}$ is either padding or end adapters (as in our case), assuming that they are of equal length on either end. As an example, for dsDNA delivered by Twist Bioscience with adapters for making a 452 bp *Circular Vector*, the maximum possible yield ratio is 0.88. Thus, if 6 μg of DNA is used as an input, the maximum possible yield will be $6 * 0.88 = 5.28$ μg.

2. We define *the molar multiplier* of the circularization reaction product as the ratio of the combined relative weight produced by circularization to the molar relative fraction of a single circular guide. We assess the relative band brightnesses from the gel electrophoresis image, which was processed using the NIH's ImageJ (see Supplementary Fig. 3 for the complete set of unmodified gel elecrtophoresis images). Since we know the relative brightness, we must sum all band relative brightnesses, which, by definition, will be = 1 divided by the sum of each band's relative brightness ($I_i$) divided by that band's number $i$. Thus, the molar multiplier $M$ is calculated as:

$$M = \frac{\sum_{i=1}^{n} I_i}{\sum_{i=1}^{n} I_i / i} = \frac{1}{\sum_{i=1}^{n} I_i / i}. \quad (2)$$

**Transfection, handling, and sequencing of edited HEK293T cells.** HEK293T cells were seeded on 24-well plates (Corning) at $140 \cdot 10^3$ cells per well in DMEM supplemented with 10% FBS. Approximately 24 h after seeding, the cells were transfected at 60% confluency with 2.0 mL of Lipofectamine 2000 in Optimem (Thermo Fisher Scientific) according to the manufacturer's protocol and 800 ng of Cas9 prime editor plasmid, with molar ratios of 4:1 (110 ng for purified single band 452 bp *Circular Vectors*, 175 ng for a 452 bp *Circular Vectors*, and 370 ng for an 952 bp *Circular Vectors*) for each of the location guides (HBB, CDKL5 and PRNP single base substitution based on[15]).

Then, 14–16 h after transfection, media was replaced with fresh DMEM plus 10% FBS and 6 ng/μl of blasticidin (Thermo Fisher Scientific) to select for cells containing prime editor. This was followed 24 h later by 6 ng/μl. Then, a media change was performed after 24 h (based on a recommendation by Xiong et al.[50]). Genomic DNA was extracted 72 h later using a Zymo Research Quick-DNA Microprep Kit (D3020).

Genomic areas of interest were amplified from 4 ng of whole genome DNA using Q5 High-Fidelity 2× Master Mix (New England BioLabs) at 24–26 cycles using the master mix protocol (see Data Availability data files for a list of the primers). This was followed by DNA separation by electrophoresis on 1% E-Gel EX Agarose Gels (Invitrogen) and the use of a Monarch DNA Gel Extraction Kit (T1020S) with one extra wash step. Sanger sequencing was performed by Azenta Genewiz using our PCR primers.

**Reagents, materials and costs.** The protocol low reagent cost is exemplified here by the breakdown of a typical 50 μl reaction. The reagents are used in quantities listed in Table 2 and Table 3. The manufacturer sources and identifiers are provided in Table 4. The bulk of the expense inherent in *Circular Vectors* is the cost of linear dsDNA ordered from a commercial supplier. We will present the costs for the example of our 452 bp *Circular Vector*. The prices we paid for the reagents were typical for retail university buyers, without any extra discounts.

**Table 4 Reagents and resources.**

| REAGENT or RESOURCE | SOURCE | IDENTIFIER |
|---|---|---|
| BsaI-HFv2 | New England BioLabs | Cat#R3733L |
| T4 DNA Ligase | New England BioLabs | Cat#M0202L |
| Adenosine-5 Triphosphate (ATP) | New England BioLabs | Cat#P0756S |
| Lipofectamine 2000 | Thermo Fisher Scientific | Cat#11668019 |
| T5 Exonuclease | New England BioLabs | Cat#M0663L |
| Tris Base (1 M, pH > 10.0) | Boston Bioproducts | Cat#BB-2686 |
| TrypLE | Thermo Fisher Scientific | Cat#12605010 |
| Puromycin Dihydrochloride | Thermo Fisher Scientific | Cat#A1113803 |
| Blasticidin S HCl | Thermo Fisher Scientific | Cat#A1113903 |
| Penicillin-Streptomycin | Thermo Fisher Scientific | Cat#15070063 |
| Gibco DMEM, high glucose, pyruvate | Thermo Fisher Scientific | Cat#11995040 |
| Opti-MEM I Reduced Serum Medium | Thermo Fisher Scientific | Cat#31985062 |
| Embryonic Stem Cell FBS | Thermo Fisher Scientific | Cat#16141002 |
| Nuclease-free H2O | Invitrogen | Cat#10977015 |
| HiFi Hot Start - Readymix | KAPA Biosystems | Cat#KK2602 |
| Q5 High-Fidelity Master Mix | New England BioLabs | Cat#M0492L |
| NEBNext Ultra II Q5 Master Mix | New England BioLabs | Cat#M0544L |
| QIAquick PCR Purification Kit | QIAGEN | Cat#28106 |
| Sodium Acetate Solution (3 M), pH 5.2 | Thermo Fisher Scientific | Cat#R1181 |
| Monarch PCR & DNA CleanUp Kit | New England BioLabs | Cat#T1030S |
| Monarch DNA Gel Extraction Kit | New England BioLabs | Cat#T1020S |
| NucleoSpin Gel and PCR Cleanup | Macherey-Nagel | Cat#74609.50 |
| QIAGEN Plasmid Plus Midi Kit | QIAGEN | Cat#12943 |
| Invitrogen 1% E-GELS EX | Invitrogen Corporation | Cat#G402021 |
| NEB 5-alpha Flq Competent *E. coli* | New England BioLabs | Cat#C2992H |
| HEK293T | ATCC | Cat#CRL-3216 |
| pCMV-PEmax-P2A-MLH1dn | Addgene | Cat#174828 |

With a circularized length of 452 bp, a 470 bp dsDNA fragment was ordered from Twist Bioscience at the list price of $0.07 per base pair, resulting in an input DNA cost of $32.90.

Usually, 1.5–2.0 µg of dsDNA fragment was delivered by Twist Bioscience. The costs of the reagents for a 50 µl protocol are presented in Supplementary Table 1, in quantities listed in Step-by-Step Protocol section. The planned output is 2.0–2.5 µg of *Circular Vector*, which requires an input of 6.0 µg of dsDNA. Such yield is sufficient for more than 10 edits on a 24-well plate and ~40 edits on a 96-well plate. The cost of this *Circular Vector* protocol is $9.30. If your supplier can deliver a sufficient quantity of DNA fragments, this is all your expense.

In our study, we had to run a PCR amplification step (Optional, priced in Supplementary Table 1), which added $4.76 to the cost. If only a small amount of *Circular Vectors* is needed, the reaction can be scaled down to a smaller quantity of available DNA, with a lower use of reagents and lower resulting cost.

**Statistics and reproducibility**. All protocol reactions were performed in independent reaction duplicates ($n = 2$) or triplicates ($n = 3$), as indicated, and yield percentage values were averaged. Gel electrophoresis of the samples above was analyzed using National Institutes of Health's ImageJ software[17] in independent sample duplicates ($n = 2$), and resulting concatemer percentage values were averaged.

The editing efficiency analysis performed from Sanger sequencing using EditR[22,23] from independent samples in triplicates ($n = 3$) or quadruplicates ($n = 4$). The $P$ values for editing efficiency calculated by EditR were $P < 5 \cdot 10^{-8}$ for all samples. The editing efficiency values were averaged, and the corresponding standard deviation values presented in Table 1.

**Reporting summary**. Further information on research design is available in the Nature Portfolio Reporting Summary linked to this article.

## Data availability

Source data are provided with this paper Supplementary Data 1. Sanger sequencing data have been deposited in GenBank database under accession numbers starting OP971846 and ending OP971880. Uncropped gel images are presented in Supplementary Fig. 3.

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

## Author contributions

R.O. designed and performed research; R.O. and G.C. analyzed data and wrote the paper.

## Competing interests

The authors declare no competing interests.
