## [Peer Review File · Communications Biology]

Reviewers' comments:

Reviewer #1 (Remarks to the Author):

The article describes how adding two type-2s sites at the end of linear DNA molecules enables their circularization via a classic one-pot restriction-ligation protocol. The efficiency for under-1kbp sequences is high enough to allow gene editing with a success rate comparable to previously used methods.

The method presented by this paper is not groundbreaking, but the fact that it compares favorably to classical preparations using full-size-plasmids is surprising, and good news, and well worth publishing. Finding a new use for a classic technique means that it will be easy to adopt and easy to automate with liquid handlers. In this optic, I enjoyed the author's focus on determining the right balance between efficiency and protocol simplicity (e.g. by proving that using the ligation product directly is not sub-optimal), and appreciated the thorough discussion of the limitations of sourcing commercial DNA (cost, error rates, inability to synthesize sequences with repeats or homologous regions) and the implications for the method.

Next are some remarks on points which I believe the paper could clarify.

I found the last sentence of the introduction "The method is not limited to gene editing" to be a possible over-statement, and did not find in the paper any example of what other applications might be. The efficiency plummeting to <10% for >1kbp sequences would prevent the use for plasmids expressing most genes, or complex promoters, or resistance markers or origin of replication.

The authors give a thorough breakdown of costs and time for their method, but to me this is only interesting if compared to the time and cost of the main competing approach, which consists in ordering circular DNA commercially. Say I have a backbone on-boarded at Twist or GeneArt (we can assume that the promoter is part of the backbone) and I order a new construct from that backbone which expresses a guide RNA. How will the cost and lead time compare to this paper's method?

One important clarification omitted by the paper is why it is important for the DNA to be circularized before transfection. Here the circularized DNA doesn't have any marker nor replication origin, and so the main advantage over linear DNA might be a better resistance to endonucleases.

Figure 3 could be an appendix figure in my opinion. Generally, the whole discussion about molar representation rests on the sentence "For transfection, it is important to use a molar proportion of the Cas9-carrying plasmid and guide plasmid". This sentence is not self-evident and I believe references are needed.

I found that the discussion of how several CRISPR guides could be added to a Circular Vector lacked of a statement on the practical limit of the number of gRNAs one could fit while still being able to circularize the vector. My understanding is that 3 or more such guide RNAs would bring the dsDNA size to over 1kb and thus impact the chances of success. (Using polycistronic gRNA may spare the need for multiple promoters in some contexts?)

Table 1 is to me the most important data in the paper (since it proves the viability of the approach). As it was obtained from limited data, it would be fair adding confidence intervals or statistical significance indications (although I understand that this would make the table less readable).

Finally, I noted these two typos:

line 135: this represents aa increase.

line 165: as describe in

Reviewer #2 (Remarks to the Author):

The paper describes a method to generate circular DNA for transfection of mammalian cells that is independent of plasmids and the need for culturing bacteria and purification of plasmid DNA from these. The example in the paper takes short synthetic DNA fragments and/or PCR fragments that encode CRISPR-related components (e.g. prime editor guide RNA expressing DNA), circularizes these via self-ligation and then demonstrates their performance as just as good as when such transfections would be done with plasmids.

The conceptual advance of self-ligating DNA into circles is not particularly novel, but there is novelty in bringing this type of protocol back in contention now that many people are transfecting large libraries of short DNA constructs - e.g. guideRNA libraries.

In general the methods, the results and the interpretation of the data were all fine. I only have 2 minor suggestions for improvements:

1. The authors should consider trying (or at least discussing) using RCA (rolling circle amplification) to improve the yields of the products. A past member of the same research team has now implemented RCA as an alternative to miniprep for obtaining large amounts of plasmid straight from colonies - see: <https://www.octant.bio/blog-posts/octopus-v2-> and <https://github.com/octantbio/octopus>
2. Some discussion is needed about whether small circles of DNA without any typical plasmid features will be able to work better or worse for transfection of mammalian cells than standard plasmids. Analysis by others on how plasmids transition to the nucleus during transfection has identified some common sequence features that can help the process - eg. the CREB binding site - <https://www.ncbi.nlm.nih.gov/pmc/articles/PMC4150871/>.

Dear Reviewer #1, we thank you for thoughtful and helpful remarks and suggestions. We modified the manuscript, and hope you will find the changes have addressed your concerns and suggestions.

1. I found the last sentence of the introduction "The method is not limited to gene editing" to be a possible over-statement, and did not find in the paper any example of what other applications might be. The efficiency plummeting to <10% for >1kbp sequences would prevent the use for plasmids expressing most genes, or complex promoters, or resistance markers or origin of replication.

You are right, we modified the sentence to state (lines 26-28): "The method's speed of preparation, low cost, and ease of use will make it a useful tool in applications requiring the expression of short RNAs and proteins."

- instead of the previous wording:

"The method is not limited to gene editing and its speed of preparation, low cost, and ease of use will make it a useful tool in a variety of applications requiring the expression of short RNAs and proteins."

2. The authors give a thorough breakdown of costs and time for their method, but to me this is only interesting if compared to the time and cost of the main competing approach, which consists in ordering circular DNA commercially. Say I have a backbone on-boarded at Twist or GeneArt (we can assume that the promoter is part of the backbone) and I order a new construct from that backbone which expresses a guide RNA. How will the cost and lead time compare to this paper's method?

Thank you, very good idea! Please see lines new in 321-322 in Discussion, and added Appendix A.8.

3. One important clarification omitted by the paper is why it is important for the DNA to be circularized before transfection. Here the circularized DNA doesn't have any marker nor replication origin, and so the main advantage over linear DNA might be a better resistance to endonucleases.

You are very much correct. While it is noted in the Methods, many people will not read that far. Agreed, it is important to explain this in the Introduction. We added lines 61-65.

4. Figure 3 could be an appendix figure in my opinion. Generally, the whole discussion about molar representation rests on the sentence "For transfection, it is important to use a molar proportion of the Cas9-carrying plasmid and guide plasmid". This sentence is not self-evident and I believe references are needed.

Regarding Figure 3 - if you find it absolutely necessary, we could move it in the Appendix. During our presentations we found it helpful for the participants to visualize that the brightness of the gel electrophoresis bands actually translates in a predominant share of the single epegRNA circular vectors, which would be lost on many readers if the figure will be moved to the Appendix. We would prefer to keep Figure 3 in the body of the paper.

The concern about molar ratio is based on experimenter's preference to have a sufficient surplus of relatively small guide expression vectors. The ratio is to make sure that each Cas9 plasmid is highly likely to have at least one or more of the small epegRNA Circular Vectors (or

corresponding expression plasmids in the classical implementation). We endeavoured to include the explanation, even though it is difficult to do so concisely; see lines 153-161.

5. I found that the discussion of how several CRISPR guides could be added to a Circular Vector lacked of a statement on the practical limit of the number of gRNAs one could fit while still being able to circularize the vector. My understanding is that 3 or more such guide RNAs would bring the dsDNA size to over 1kb and thus impact the chances of success. (Using polycistronic gRNA may spare the need for multiple promoters in some contexts?)

We thought that polycistronic gRNA is an obvious simple use of the method. Thus, we were aiming to give an example requiring additional assembly. However, it is a good idea to mention polycistronic gRNA, as it makes such implementation options clear, please see added lines 277-285. Please note that the sentences in purple in the middle of this addition are answering the reviewer #2 suggestion, which seemed fitting to combine here.

6. Table 1 is to me the most important data in the paper (since it proves the viability of the approach). As it was obtained from limited data, it would be fair adding confidence intervals or statistical significance indications (although I understand that this would make the table less readable).

Certainly, good point to add confidence intervals. Also, it gave a chance to calculate precise values for the reference paper editing efficiency values. Table 1 now includes the mean and SD. It took a slight rearrangement, and the table remains nicely readable.

**7. Finally, I noted these two typos:
line 135: this represents aa increase.
line 165: as describe in**

Thank you, corrected.

Dear Reviewer #2, we thank you for thoughtful review, remarks and suggestions. We modified the manuscript, and hope you will find the changes have addressed your concerns and suggestions.

1. The authors should consider trying (or at least discussing) using RCA (rolling circle amplification) to improve the yields of the products. A past member of the same research team has now implemented RCA as an alternative to miniprep for obtaining large amounts of plasmid straight from colonies - see: <https://www.octant.bio/blog-posts/octopus-v2-> and <https://github.com/octantbio/octopus>

This is a very interesting thought about using RCA. However, we wish to kindly explain why adding discussion of RCA as a synthesis method would be out of scope of this research:

a) The RCA is primarily used for producing long linear concatemer repeat sequences, unless special steps are taken for splicing and circularization. We aimed at developing the simplest possible method for preparation of such circular structures. It would be entirely different research project to accomplish this.

b) There is no issue with the yield for the intended purpose, as our reaction yield proven to be sufficient for 10-20 editing runs, which was always a few times more than needed for our genome editing purposes; if needed, scaling up our reaction can easily produce more product.

However, we find the Octant RCA sequencing method interesting and worth noting, see added lines 370-374.

2. Some discussion is needed about whether small circles of DNA without any typical plasmid features will be able to work better or worse for transfection of mammalian cells than standard plasmids. Analysis by others on how plasmids transition to the nucleus during transfection has identified some common sequence features that can help the process - eg. the CREB binding site - <https://www.ncbi.nlm.nih.gov/pmc/articles/PMC4150871/>.

Certainly, it is worth discussing, and this question coincides nicely with reviewer #1 slightly different question, thus both suggestions are discussed within the same paragraph. See added lines 278-282.

Reviewer #1 (Remarks to the Author):

The article describes how adding two type-2s sites at the end of linear DNA molecules enables their circularization via a classic one-pot restriction-ligation protocol. The efficiency for under-1kbp sequences is high enough to allow gene editing with a success rate comparable to previously used methods.

The method presented by this paper is not groundbreaking, but the fact that it compares favorably to classical preparations using full-size-plasmids is surprising, and good news, and well worth publishing. Finding a new use for a classic technique means that it will be easy to adopt and easy to automate with liquid handlers. In this optic, I enjoyed the author's focus on determining the right balance between efficiency and protocol simplicity (e.g. by proving that using the ligation product directly is not sub-optimal), and appreciated the thorough discussion of the limitations of sourcing commercial DNA (cost, error rates, inability to synthesize sequences with repeats or homologous regions) and the implications for the method.

Next are some remarks on points which I believe the paper could clarify.

Dear Reviewer #1, we thank you for thoughtful and helpful remarks and suggestions. We modified the manuscript, and hope you will find the changes have addressed your concerns and suggestions.

1. I found the last sentence of the introduction "The method is not limited to gene editing" to be a possible over-statement, and did not find in the paper any example of what other applications might be. The efficiency plummeting to <10% for >1kbp sequences would prevent the use for plasmids expressing most genes, or complex promoters, or resistance markers or origin of replication.

You are right, we modified the sentence to state (lines 26-28): "The method's speed of preparation, low cost, and ease of use will make it a useful tool in applications requiring the expression of short RNAs and proteins."

- instead of the previous wording:

"The method is not limited to gene editing and its speed of preparation, low cost, and ease of use will make it a useful tool in a variety of applications requiring the expression of short RNAs and proteins."

2. The authors give a thorough breakdown of costs and time for their method, but to me this is only interesting if compared to the time and cost of the main competing approach, which consists in ordering circular DNA commercially. Say I have a backbone on-boarded at Twist or GeneArt (we can assume that the promoter is part of the backbone) and I order a new construct from that backbone which expresses a guide RNA. How will the cost and lead time compare to this paper's method?

Thank you, very good idea! Please see lines new in 321-322 in Discussion, and added Appendix A.8.

3. One important clarification omitted by the paper is why it is important for the DNA to be circularized before transfection. Here the circularized DNA doesn't have any marker

nor replication origin, and so the main advantage over linear DNA might be a better resistance to endonucleases.

You are very much correct. While it is noted in the Methods, many people will not read that far. Agreed, it is important to explain this in the Introduction. We added lines 61-65.

4. Figure 3 could be an appendix figure in my opinion. Generally, the whole discussion about molar representation rests on the sentence "For transfection, it is important to use a molar proportion of the Cas9-carrying plasmid and guide plasmid". This sentence is not self-evident and I believe references are needed.

Regarding Figure 3 - if you find it absolutely necessary, we could move it in the Appendix. During our presentations we found it helpful for the participants to visualize that the brightness of the gel electrophoresis bands actually translates in a predominant share of the single epegRNA circular vectors, which would be lost on many readers if the figure will be moved to the Appendix. We would prefer to keep Figure 3 in the body of the paper.

The concern about molar ratio is based on experimenter's preference to have a sufficient surplus of relatively small guide expression vectors. The ratio is to make sure that each Cas9 plasmid is highly likely to have at least one or more of the small epegRNA Circular Vectors (or corresponding expression plasmids in the classical implementation). We endeavoured to include the explanation, even though it is difficult to do so concisely; see lines 153-161.

5. I found that the discussion of how several CRISPR guides could be added to a Circular Vector lacked of a statement on the practical limit of the number of gRNAs one could fit while still being able to circularize the vector. My understanding is that 3 or more such guide RNAs would bring the dsDNA size to over 1kb and thus impact the chances of success. (Using polycistronic gRNA may spare the need for multiple promoters in some contexts?)

We thought that polycistronic gRNA is an obvious simple use of the method. Thus, we were aiming to give an example requiring additional assembly. However, it is a good idea to mention polycistronic gRNA, as it makes such implementation options clear, please see added lines 277-285. Please note that the sentences in purple in the middle of this addition are answering the reviewer #2 suggestion, which seemed fitting to combine here.

6. Table 1 is to me the most important data in the paper (since it proves the viability of the approach). As it was obtained from limited data, it would be fair adding confidence intervals or statistical significance indications (although I understand that this would make the table less readable).

Certainly, good point to add confidence intervals. Also, it gave a chance to calculate precise values for the reference paper editing efficiency values. Table 1 now includes the mean and SD. It took a slight rearrangement, and the table remains nicely readable.

**7. Finally, I noted these two typos:
line 135: this represents aa increase.
line 165: as describe in**

Thank you, corrected.

Reviewer #2 (Remarks to the Author):

The paper describes a method to generate circular DNA for transfection of mammalian cells that is independent of plasmids and the need for culturing bacteria and purification of plasmid DNA from these. The example in the paper takes short synthetic DNA fragments and/or PCR fragments that encode CRISPR-related components (e.g. prime editor guide RNA expressing DNA), circularizes these via self-ligation and then demonstrate their performance as just as good as when such transfections would be done with plasmids.

The conceptual advance of self-ligating DNA into circles is not particularly novel, but there is novelty in bringing this type of protocol back in contention now that many people are transfecting large libraries of short DNA constructs - e.g. guideRNA libraries.

In general the methods, the results and the interpretation of the data were all fine. I only have 2 minor suggestions for improvements:

Dear Reviewer #2, we thank you for thoughtful review, remarks and suggestions. We modified the manuscript, and hope you will find the changes have addressed your concerns and suggestions.

1. The authors should consider trying (or at least discussing) using RCA (rolling circle amplification) to improve the yields of the products. A past member of the same research team has now implemented RCA as an alternative to miniprep for obtaining large amounts of plasmid straight from colonies - see: <https://www.octant.bio/blog-posts/octopus-v2-> and <https://github.com/octantbio/octopus>

This is a very interesting thought about using RCA. However, we wish to kindly explain why adding discussion of RCA as a synthesis method would be out of scope of this research:

- a) The RCA is primarily used for producing long linear concatemer repeat sequences, unless special steps are taken for splicing and circularization. We aimed at developing the simplest possible method for preparation of such circular structures. It would be entirely different research project to accomplish this.
- b) There is no issue with the yield for the intended purpose, as our reaction yield proven to be sufficient for 10-20 editing runs, which was always a few times more than needed for our genome editing purposes; if needed, scaling up our reaction can easily produce more product.

However, we find the Octant RCA sequencing method interesting and worth noting, see added lines 370-374.

2. Some discussion is needed about whether small circles of DNA without any typical plasmid features will be able to work better or worse for transfection of mammalian cells than standard plasmids. Analysis by others on how plasmids transition to the nucleus during transfection has identified some common sequence features that can help the process - eg. the CREB binding site - <https://www.ncbi.nlm.nih.gov/pmc/articles/PMC4150871/>.

Certainly, it is worth discussing, and this question coincides nicely with reviewer #1 slightly different question, thus both suggestions are discussed within the same paragraph. See added lines 278-282.